# A Comprehensive Functional Analysis of *OsPEAMT1* and *OsPEAMT2* Genes in Rice (*Oryza sativa* L. ssp. *japonica*)

**DOI:** 10.3390/plants14182935

**Published:** 2025-09-22

**Authors:** Jinde Yu, Yuying Zhang, Shaojie Ma, Xia Wen, Ning Zhao, Xiaofei Feng, Dan Zong, Jing Li

**Affiliations:** 1College of Biological Science and Food Engineering, Southwest Forestry University, Kunming 650224, China; 2Vocational Education Research Center, Yunnan Vocational College of Agriculture, Kunming 650201, China

**Keywords:** phosphoethanolamine *N*-methyltransferase, phosphocholine, phosphatidylcholine, recombinant protein, expression patterns, stress, rice

## Abstract

Phosphoethanolamine *N*-methyltransferase (PEAMT) is a key enzyme that catalyzes three successive methylation steps of phosphoethanolamine (P-EA) to phosphocholine (P-Cho). Meanwhile, P-Cho is a major precursor of phosphatidylcholine (PC) and glycine betaine (GB), which are involved in cell signal transduction, stress response, etc. Therefore, the *PEAMT* gene plays an essential role in plant growth and development as well as stress resistance. There are two homologous *PEAMT* genes in rice (*Oryza sativa* L. ssp. *japonica*), namely, *OsPEAMT1* and *OsPEAMT2.* However, there has not been any comprehensive functional analysis of these two genes. Here, we employed bioinformatics methods to analyze the amino acid sequences and promoters of *OsPEAMT1* and *OsPEAMT2*, and found that both proteins contain two methyltransferase domains. OsPEAMT1 is highly similar with ZmPEAMT, and OsPEAMT2 is closely related to LmPEAMT and TaPEAMT. There are abundant plant hormone response elements, stress response elements and low-temperature response elements in the promoters of *OsPEAMT1* and *OsPEAM*T2. The in vitro enzymatic activity assays of recombinant proteins of OsPEAMT1 and OsPEAMT2 indicated that they can catalyze the production of P-Cho from P-EA, respectively. Meanwhile, the endogenous P-Cho content increased significantly (*p* < 0.05) when exogenous P-EA was added to rice. These indicate that OsPEAMT1 and OsPEAMT2 proteins have catalytic functions in vivo and in vitro. The expression patterns of both genes are different in different tissues, flowers and seeds at various developmental stages. Additionally, both genes have different responses to salt and low-temperature stress. This study supplies valuable insights into the function of *OsPEAMT1* and *OsPEAMT2*, and it will provide key targets for rice molecular breeding, offering important insights for the development of rice with stress resistance and high yield.

## 1. Introduction

*PEAMT*, the encoding gene of phosphoethanolamine *N*-methyltransferase (PEAMT), catalyzes the three sequential methylations (S-adenosyl-L-methionine as the methyl donor) of phosphoethanolamine (P-EA) to form phosphomonomethylethano-lamine (PMMEA) and phosphodimethylethanolamine (PDMEA) and finally produce phosphocholine (P-Cho) [1]. P-Cho is an important precursor of phosphatidylcholine (PC) and glycine betaine (GB), in which PC is the most abundant phospholipid, accounting for 40 to 60% of lipids in plant membranes, and it can be converted into signaling molecules such as phosphatidic acid and diacylglycerols [2,3]. For GB, it is an important cell osmotic regulator when plants are exposed to abiotic stresses such as salinity, drought, low temperatures, etc. [4,5,6]. Thus, the *PEAMT* gene plays a critical role in plant growth and development as well as abiotic stress.

Due to its importance, *PEAMT* has been successively cloned in Arabidopsis (*Arabidopsis thaliana*), spinach (*Spinacia oleracea*), wheat (*Triticum aestivum*), maize (*Zea mays*), etc. [7,8,9,10]. BeGora et al. found that the N-terminal and C-terminal of the PEAMT enzyme each contain one S-adenosine-L-methionine-dependent methyltransferase domain, and each domain contains four motifs (I, p-I, II, and III), which play a major catalytic role [11]. Meanwhile, the integrity of the N-terminal and C-terminal domains is critical to the catalytic function of the PEAMT enzyme [12]. In the Arabidopsis genome, there are three genes encoding PEAMT: AtPEAMT1/NMT1, AtPEAMT2/NMT2, and AtPEAMT3/NMT3 [11,12,13]. When NMT1 was knocked out, the nmt1 mutant showed the overt phenotypes, including temperature-sensitive male sterility, a shorter primary root, epidermal cell death, and increased lateral root numbers [2,11,12,13,14]. Meanwhile, NMT1 is important to maintain the Arabidopsis root apical meristem by affecting auxin regulation, ABA signaling, and reactive oxygen species (ROS) [15,16]. However, when NMT2 or NMT3 was knocked out, the mutant had no differential phenotype compared with the wild type [11,12]. Subsequently, one study showed that the nmt1 nmt3 double mutant has extensive sterility and severe defects in the aerial organs and root [5]. Some studies have shown that the PEAMT gene is also closely related to abiotic stress in plants. For example, the CsPEAMT expression of Cleistogenes songorica increased largely during drought stress in root and leaf [17]. Meanwhile, overexpression of the pitaya (*Hylocereus polyrhizus*) HpPEAMT in tobacco (*Nicotiana tabacum*) enhanced the drought tolerance and antioxidant defense capacity of the transgenic plants [18]. Under salt stress, the mRNA level of SoPEAMT in *Spinacia oleracea* increased approximately tenfold [8], and the expression of the SdPEAMT in the halophyte Sueda dendroides transiently surged more than fivefold [19]. Similarly, the expression of BnPEAMT was significantly increased when rape (*Brassica napus*) was under salinity stress [20].

In addition, soil bacterium *Bacillus subtilis* (GB03) co-cultured with Arabidopsis can enhance AtPEAMT activity greatly, thus enhancing the tolerance of Arabidopsis to drought stress [21]. Adak et al. showed that the beneficial rhizobacteria can induce the expression of *PEAMT* in the host plant through IAA signaling, promoting the rearrangement of the membrane lipid and stabilizing the cell membrane, thereby enhancing the plant’s salt tolerance [22]. Among crops, when wheat was exposed to low-temperature stress, the TaPEAMT activity increased significantly, and it can synergize with *TaCBF14* (cold-resistant gene) to improve cell membrane stability [23,24]. Meanwhile, a study of the promoter of the maize *PEAMT* gene revealed that *ZmPEAMT* has a positive response to abiotic stress such as salt, drought, and cold stress [25]. Furthermore, four days after maize pollination, the expression of *ZmPEAMT1* was significantly upregulated, consequently providing sufficient P-Cho for the developing kernels and enhancing cell division and elongation, thereby promoting rapid growth of the early ear [26].

Rice (*Oryza sativa*) is one of the most important food crops that feeds more than half of the world’s population [27]. However, it is highly sensitive to abiotic stresses such as salinity and low temperature; therefore, these abiotic stresses are significant factors contributing to the reduction in rice yield and quality [28]. Previous studies have demonstrated that the *PEAMT* gene confers abiotic stress tolerance in other plant species, but there has been little research on the rice *PEAMT* family [5,11,12]. In the rice cv. Nipponbare (*Oryza sativa* L. ssp. *japonica*) genome, there are two homologous *PEAMT* genes: *OsPEAMT1* (Os01g50030) and *OsPEAMT2* (Os05g47540) [29]. Here, in order to comprehensively investigate the functions of *OsPEAMT1* and *OsPEAMT2*, we primarily analyzed the *cis*-acting elements for abiotic stress resistance of *OsPEAMT1* and *OsPEAMT2* promoters, and the functional domains of the OsPEAMT1 and OsPEAMT2 proteins were predicted by sequence and phylogenetic analysis. Subsequently, we determined the catalytic activity of the OsPEAMT1 and OsPEAMT2 proteins in vivo and in vitro, respectively. Meanwhile, the expression patterns of *OsPEAMT1* and *OsPEAMT2* and the content of P-Cho were determined in different tissues, flowers, and seeds at various developmental stages. Additionally, we revealed that *OsPEAMT1* and *OsPEAMT2* respond to salt and low-temperature stress differently. Subsequently, we found that exogenous P-Cho can increase the resistance of rice to salt and low-temperature stress. Together, our results provide a comprehensive understanding of the function of *OsPEAMT1* and *OsPEAMT2*, and they can also provide valuable genetic resources for rice molecular breeding.

## 2. Results

### 2.1. Identification of OsPEAMT1 and OsPEAMT2

Through filtering and domain analysis, both *OsPEAMT1* and *OsPEAMT2* contain 12 exons and 11 introns. The CDS of *OsPEAMT1* has 1500 bp and encodes a predicted protein (499 amino acids) with an isoelectric point (pI) of 5.4 and a putative molecular mass of 54.89 kDa, and the CDS of *OsPEAMT2* has 1488 bp and encodes a predicted protein (495 amino acids) with a pI of 5.8 and a putative molecular mass of 54.45 kDa. The N-terminal and the C-terminal of OsPEAMT1 and OsPEAMT2 have a methyltransferase (MT) domain, respectively (Figure 1A). Each domain has four SAM-binding motifs, namely I, p-I, II, and III. The alignment analysis shows that the domains of MT1 and MT2 of OsPEAMT1 and OsPEAMT2 are homologous to other plant PEAMTs (Figure 1B,C). This result indicates that *OsPEAMT1* and *OsPEAMT2* probably encode the functional OsPEAMT proteins.

To investigate the phylogenetic relationships of OsPEAMT1 and OsPEAMT2 with other plant PEAMTs, the neighbor-joining tree method was used to construct a phylogenetic tree (Figure 2). These PEAMT proteins are derived from monocotyledon and dicotyledon. The result shown that OsPEAMT1 is highly similar with *Zea mays* ZmPEAMT; however, OsPEAMT2 is closely related to *Leymus mollis* LmPEAMT and *Triticum aestivum* TaPEAMT, hinting that they may have similar functions.

### 2.2. Identification of OsPEAMT1 and OsPEAMT2 Promoters

The 2000 bp upstream of start codon of *OsPEAMT1* and *OsPEAMT2* was isolated, and the potential *cis*-acting elements of *OsPEAMT1* and *OsPEAMT2* promoters were predicted by PlantCARE (Figure 3). Abundant response elements of plant hormones (abscisic acid, gibberellin, salicylic acid, etc.) and light were found in the region of *OsPEAMT1* and *OsPEAM*T2 promoters. Meanwhile, there are some stress response elements (STRE) in the *OsPEAMT1* promoter not found in the *OsPEAM*T2 promoter. Notably, both promoters of *OsPEAMT1* and *OsPEAMT2* contain a low-temperature response element (LTR) and MYB-binding site (MBS). The above predictive analysis suggests that both genes play significant biological roles in plant hormone responses, light signaling, and abiotic stress adaptation. The details of the *cis*-acting elements of *OsPEAMT1* and *OsPEAM*T2 promoters are showed in Appendix A.

### 2.3. Detection of OsPEAMT1 and OsPEAMT2 Activity

To determine whether the OsPEAMT1 and OsPEAMT2 enzymes can catalyze P-EA to produce P-Cho, respectively, the CDSs of *OsPEAMT1* and *OsPEAMT2* were expressed in BL21 using a pET30a vector, respectively. The recombinant proteins were induced by 0.2 mM IPTG and were purified on a His-bind column (Appendix A). The target proteins were dialyzed overnight and detected by SDS-PAGE electrophoresis. The results showed that the higher quality purified OsPEAMT1 (0.51 mg/mL) and OsPEAMT2 (0.55 mg/mL) recombinant proteins were obtained through prokaryotic expression, respectively (Figure 4). To detect the enzymes activity, the substrate P-EA was added to the reaction solution containing OsPEAMT1 and OsPEAMT2 enzymes (as described in Materials and Methods), respectively. The result shows that both recombinant OsPEAMT1 and OsPEAMT2 have the activity to catalyze substrate P-EA producing P-Cho (the retention time for P-Cho determination by LC-MS/MS was 0.840 min, Appendix A), but the activity of OsPEAMT1 was significantly higher than that of OsPEAMT2 (*p* < 0.05) (Figure 4).

To further test the activity of OsPEAMT1 and OsPEAMT2 enzymes in vivo, exogenous substrate P-EA (0, 10, 20, and 40 mM) was supplied to 3-week-old rice seedlings, respectively. We found that the content of the product P-Cho increased significantly with increasing concentrations of the applied exogenous substrate P-EA (Figure 5). Namely, in the absence of exogenous P-EA (0 mM) supplementation, the product P-Cho was 107.1 ng/mg (fresh weight, FW); however, after P-EA was applied at concentrations of 10, 20, and 40 mM, the P-Cho levels significantly reached 144.9, 177.7, and 206.3 ng/mg, respectively. These results further indicate that as the concentration of the P-EA increases, the enzymatic reaction rate of OsPEAMTs accelerates, and the accumulation of P-Cho increases significantly. Furthermore, the increasing trend in P-Cho suggests that the 40 mM substrate P-EA may not have reached saturation for the OsPEAMT enzyme. In any case, these results imply that at least one or both OsPEAMT1 and OsPEAMT2 can catalyze P-EA to produce P-Cho in rice.

### 2.4. Expression Analysis of OsPEAMT1 and OsPEAMT2 in Different Tissues, Flowers, and Seeds at Various Developmental Stages

RT-qPCR was used to detect the expression ratio of *OsPEAMT1* and *OsPEAMT2* in different tissues (root, stem, leaf), flowers, and seeds at various developmental stages. As shown in Figure 6, the transcripts of both *OsPEAMT1* and *OsPEAMT2* can be detected in roots, stems, leaves, and the flowers and seeds at various developmental stages, but their expression patterns are not exactly the same. *OsPEAMT1* has high expression levels in leaves, late stages of flower development (6 and 8 days), and seed development stages, but low expression levels in roots, stems, and early stages of flower development (2 and 4 days) (Figure 6A). *OsPEAMT2* has high expression levels in leaves, early stages of flower development and stems, but low expression levels in roots, late stages of flower development, and seed development stages (Figure 6B). However, the expression level of *OsPEAMT1* gradually increased during flower development (Figure 6A), while *OsPEAMT2* gradually decreased (Figure 6B). This indicates that *OsPEAMT1* and *OsPEAMT2* functionally complement each other at the flower development stages. Nevertheless, both of their expression levels increased gradually at the seed development stages.

### 2.5. Determination of P-Cho Content in Different Tissues, Flowers, and Seeds at Various Developmental Stages

Since P-Cho is the catalytic product of PEAMT enzyme, we also determined the content of P-Cho in tissues, flowers, and seeds at various developmental stages using LC-MS/MS. P-Cho can be detected in roots, stems, leaves, flowers, and seeds at various developmental stages (Figure 7). However, except for the 8-day stage of flower development, the content of P-Cho in the other stages of flower development was significantly higher than that in other tissue and seed development stages. Meanwhile, the content of P-Cho in roots was significantly lower than that in other tissues, flowers, and seeds at various developmental stages. In addition, with the development of flowers and seeds, the content of P-Cho decreased gradually. Nevertheless, the variation in P-Cho content in different tissues, particularly during seed development stages, is not consistently aligned with the expression trends of *OsPEAMT1* or *OsPEAMT2* (Figure 6). Previous studies have demonstrated that the downstream products of P-Cho, such as choline, PC, and acetylcholine, are highly abundant in cereal seeds [30,31,32]. Accordingly, this inconsistent trend between the P-Cho content and *OsPEAMT1* expression might be attributed to the inhibition of P-Cho synthesis or its conversion to other metabolites, such as choline and PC.

### 2.6. Expression Analysis of OsPEAMT1 and OsPEAMT2 Under Salt or Low-Temperature Stress

We further investigated whether the expression of *OsPEAMT1* and *OsPEAMT2* is induced by salt or low-temperature stress. As shown in Figure 8, the expression of *OsPEAMT1* was not affected under different concentrations of salt (0, 50, 100, and 150 mM) stress, but it increased more than fourfold under low-temperature (10 °C) stress (Figure 8A). However, under salt stress conditions of 50, 100, and 150 mM, the relative expression levels of *OsPEAMT2* decreased to 0.47, 0.26, and 0.11, respectively, and under low-temperature stress, its relative expression levels decreased to 0.21 (Figure 8B). These results indicate that the response patterns of *OsPEAMT1* and *OsPEAMT2* to salt or low-temperature stress are different. *OsPEAMT1* positively responded to low-temperature stress, but the transcription level of *OsPEAMT2* decreased in response to salt and low-temperature stress.

### 2.7. Response Analysis of Exogenous P-Cho to Salt and Low-Temperature Stress in Rice

Since P-Cho is the catalytic product of OsPEAMT1 and OsPEAMT2, we also determined whether exogenous P-Cho can enhance rice resistance to salt and low-temperature stress. When 0, 50, 100, and 150 mM of exogenous P-Cho were supplied to 3-week-old rice seedlings, these seedlings showed increased resistance to salt stress (150 mM) (Figure 9A) and low-temperature (10 °C) (Figure 9B) stress as the concentration of exogenous P-Cho increased. Furthermore, we determined the root length, the number of roots, the seedling length, and the fresh weight of these seedlings (Figure 9C–F). The results show that, under 150 mM NaCl or 10 °C treatment, the root length (Figure 9C), seedling length (Figure 9E), and fresh weight (Figure 9F) of rice seedlings were significantly higher at high concentrations of P-Cho than at low concentrations. But there was no significant difference in the number of roots at any concentration of P-Cho. These results indicate that P-Cho can enhance the resistance of rice to salt and low-temperature stress.

## 3. Discussion

Studies in other plants have demonstrated that the *PEAMT* gene plays a vital role in plant responses to various environmental stresses as well as in growth and developmental processes, including root growth and flower formation [5,11,12,16,18,28]. This study conducted a comprehensive investigation into the functions of the *PEAMT* gene family, namely *OsPEAMT1* and *OsPEAMT2*, in rice. The bioinformatic analyses and in vitro and in vivo experiments demonstrated that both OsPEAMT1 and OsPEAMT2 proteins have the capacity to catalyze P-EA into P-Cho, respectively (Figure 1, Figure 2, Figure 4 and Figure 5). Phylogenetic analyses of plant PEAMTs show that OsPEAMT1 is highly similar to the functional ZmPEAMT; however, OsPEAMT2 is closely related to the functional LmPEAMT and TaPEAMT (Figure 2). This indicates that OsPEAMT1 and OsPEAMT2 share functional similarities with ZmPEAMT, LmPEAMT, and TaPEAMT, respectively [10,23]. Bioinformatic analysis revealed that both OsPEAMT1 and OsPEAMT2 contain an N-terminal and a C-terminal methyltransferase domain (MT) (Figure 1). However, it remains unclear whether these two MTs can catalyze all three methylation steps or only partial methylation steps. Previous studies have shown that the absence of the C-terminal MT of PEAMT of spinach results in the N-terminal MT domain only catalyzing the first methylation step, while the C-terminal MT may catalyze the latter two steps [8,33]. In contrast, in wheat, the absence of the N-terminal MT leads to a complete loss of methyltransferase activity, whereas the absence of the C-terminal MT has no effect on methyltransferase activity [9]. These suggest that the catalytic properties of the N-terminal and C-terminal MT of PEAMT vary across plant species. Therefore, the differential roles of the N-terminal and C-terminal MT in catalyzing methylation reactions in OsPEAMT1 and OsPEAMT2 will be an intriguing scientific question in our subsequent research.

The acquisition of functional diversity among closely related isoforms within gene families commonly occurs through alterations of expression patterns [4]. Consistent with this, the *OsPEAMT1* and *OsPEAMT2* genes present partly overlapping but also specific expression patterns in flower development stages; for example, although both *OsPEAMT1* and *OsPEAMT2* were expressed at the highest levels in leaves as well as having similar expression patterns at the seed development stages, the expression patterns of *OsPEAMT1* and *OsPEAMT2* are the opposite in flower development stages (Figure 6). These results suggest that *OsPEAMT1* and *OsPEAMT2* are functional homologs but they are not totally redundant, and even complement each other. In Arabidopsis, there are three *PEAMT* homologous genes, but their expression patterns are not identical in reproductive development [4]. Meanwhile, there are two *PEAMT* homologous genes in soybean (*Glycine max*), and their expression patterns are different in different tissues [30]. These findings are consistent with the results of our study.

P-Cho is the catalytic product of the PEAMT enzyme in plants; however, the content of P-Cho is inconsistent with the expression levels of *OsPEAMT1* and *OsPEAMT2* in our study (Figure 6 and Figure 7). This contradiction partly accounts for P-Cho, which can also be further converted into substances such as PC and GB by corresponding enzymes [30,31,32]. Studies have shown that P-Cho can couple to cytidine-5′ triphosphate (CTP) to form cytidine diphosphate (CDP) choline through a reaction catalyzed by choline-phosphate cytidyltransferase, and the choline moiety of CDP-choline is transferred to diacylglycerols (DAGs) by CDP-choline 1,2-diacylglycerol-cholinephosphotransferase to produce PC [4]. Then PC can serve as a major precursor of storage lipid biosynthesis and lipid-based signaling molecules, such as (DAGs) and phosphatidic acid (PA), which are involved in stress responses in plants [3]. Furthermore, during plant growth and development or under abiotic stress conditions, the choline moiety in P-Cho can be catalyzed by choline monooxygenase (CMO) to form betaine aldehyde, which is then converted to GB by the action of betaine aldehyde dehydrogenase (BADH) [29,33]. Therefore, although *OsPEAMT1* and *OsPEAMT2* expression is the highest in the leaves, P-Cho may be further converted into these substances in our study. Similarly, the expression levels of *OsPEAMT1* and *OsPEAMT2* increased gradually in the seed development stages, but the content of P-Cho did not increase correspondingly, and even decreased. Previous studies have shown that plant seeds are rich in PC [34]; therefore, it is possible that the P-Cho was converted to PC during the seed development of rice. It is also worth noting that the P-Cho transporter may influence the content of P-Cho in different tissues [4], which requires further exploration.

When plants encounter abiotic stress, the expression of corresponding resistance genes can be induced, thereby reducing the harm caused by the stress to plants [9]. And this induced expression is strictly regulated by the *cis*-acting elements in the upstream promoter region of the genes [35]. In this work, we found that *OsPEAMT1* expression was not induced by salt, but its expression sharply increased at a low temperature (10 °C) (Figure 8A). We found that the promoter region of *OsPEAMT1* has low-temperature inducible elements, but no obvious salt-inducing elements (Figure 3, Appendix A). Meanwhile, in wheat and soybean, a low temperature can also upregulate *TsPEAMT* and *GsPEAMT* expression [9,33]. These results are consistent with the findings in our study. It is interesting that under salt and low-temperature (10 °C) conditions, the transcription level of *OsPEAMT2* showed a decreasing trend (Figure 8B); this phenomenon could be the result of the suppression of *OsPEAMT2* transcription by stress-related factors. For example, under salt stress, the transcription factor OfWRKY157 specifically binds to the W-box (TTGACC) *cis*-element in the *OfSOD1* promoter and represses its transcription, resulting in reduced ROS scavenging capacity and consequently enhancing the sensitivity of *Osmanthus fragrans* to salt stress [36]. Additionally, the transcription factor OsSGL directly suppresses the transcription of *OsNCED3*, a key gene in ABA biosynthesis, thereby negatively regulating salt tolerance in rice [37]. However, through the observation of the promoters, we found that the elements in the promoter of the *OsPEAMT1* and *OsPEAMT2* were not identical (Figure 3, Appendix A). This difference may affect the binding of their transcription factors to their promoters when rice was under low-temperature or salt stress: this point is worthwhile to study further.

P-Cho is not only the product of OsPEAMT1 and OsPEAMT2, it is also a precursor of the signaling molecules, including PC, DAGs, and PA, which are involved in stress responses in plants [38,39]. In our study, we found that the application of exogenous P-Cho to rice increased the resistance of rice to salt and low-temperature stress, and the resistance increased with the increase in P-Cho concentration (Figure 9). This may be due to the further conversion of exogenous P-Cho into PC or other correlated signaling molecules in rice. In addition, it has been shown that PC is an important component of the plant cell membrane, which can enhance plant resistance to abiotic stress, including low-temperature and high salinity [40]. Thus, the application of exogenous P-Cho can not only reduce the cost of agronomic management, such as mitigating expenses associated with abiotic stress damage to crops, but can also reduce the workforce of agronomic management in crop cultivation. Moreover, investigating the molecular mechanisms of phosphocholine (P-Cho) in stress resistance and identifying stress-responsive genes can provide excellent genetic resources for the molecular design breeding of rice. In short, the results of this study suggest that *OsPEAMT1* and *OsPEAMT2* play an important role in resistance to abiotic stress.

## 4. Materials and Methods

### 4.1. Plant Materials and Growth Conditions

Rice (*Oryza sativa* L. ssp. *japonica*) seeds were stored at 4 °C in our laboratory. For seedlings grown in a tissue culture bottle, seeds were sterilized with 75% (*v*/*v*) ethanol for 2 min, then sterilized with 25% [*v*/*v*] sodium hypochlorite for 18 min, washed five times with sterile water, and dried. The sterilized seeds were sown in a bottle with two layers of filter paper at the bottom, 30 seeds per bottle, with 20 mL of sterile water. Bottles were placed in a growth chamber set at 25 °C for 16 h of light/8 h of dark, unless specified otherwise. For rice plants grown in soil, they were cultivated in a field (Yuanjiang County, Yunnan, China). The planting density was 15 plants/m^2^, and during the field cultivation period (June 2024–October 2024), the average daily light exposure was 12.8 h, with a relative humidity of 67.3%.

### 4.2. Analysis of OsPEAMT1 and OsPEAMT2 Promoters

The genome assembly sequence of rice cv. Nipponbare [41] was downloaded from the National Center for Biotechnology Information (NCBI), and the promoter regions of *OsPEAMT1* and *OsPEAMT2* (2000 bp upstream of the gene start codon) were extracted by Python script. The *cis*-acting elements of *OsPEAMT1* and *OsPEAMT2* promoters were predicted by PlantCARE [42] and visualized with TBtools v1.132 [43].

### 4.3. Sequence and Phylogenetic Analysis of OsPEAMT1 and OsPEAMT2 Proteins

The PEAMT proteins of rice, Arabidopsis, maize, and other plant species were downloaded from NCBI. Multiple amino acid sequence alignments of PEAMT proteins were performed using ClustaW with default parameters [44]. The phylogenetic tree was constructed in MEGA 11 using the neighbor-joining (NJ) method, and the parameters of pair-wise deletion, Poisson correction, and 1000 bootstrap replicates [45]. Through the results of multiple sequence alignment, the conserved domains of methyltransferase (MT) of PEAMT proteins were visualized using Jalview Version 2 [46], and the schematic diagrams of domains were drawn using Illustrator for Biological Sequences (IBS) [47].

### 4.4. Expression and Purification of OsPEAMT1 and OsPEAMT2 Proteins

The CDSs of *OsPEAMT1* and *OsPEAMT2* were amplified by PCR from the rice cDNA library, respectively. And *OsPEAMT1* was amplified using primers 5′-CCCCTCTAAAATCGCACTGA-3′ and 5′-AATTTATTTGCGAGTCCCCC-3′, while *OsPEAMT2* was amplified using primers 5′-TAGACGAGGACGAGGAGAC-3′ and 5′-ACAGCAGCACCCGTTTTTATTC-3′. The PCR products of *OsPEAMT1* and *OsPEAMT2* were cloned into the pEASY-Blunt vector (TransGen, Beijing, China), respectively. And the positive plasmids were sequenced by Yingjun Biotechnology (Shanghai, China). The sequencing results were compared with the database, and the codons of *OsPEAMT1* and *OsPEAMT2* were optimized for expression in *Escherichia coli* (*E. coli*) using online tool ExpOptimizer “https://www.novopro.cn/tools/codon-optimization.html (accessed on 28 March 2024)”. We synthesized the optimized CDSs of *OsPEAMT1* and *OsPEAMT2* by chemical methods at Wuhan Biorun Biosciences Co., Ltd. (Wuhan, China). The synthesized CDSs for *OsPEAMT1* and *OsPEAMT2* were amplified by PCR using the following primers: for *OsPEAMT1*, 5′-TTAAGAAGGAGATATACATAATGGCACAGAAAAGCTACTGGGAAGAAC-3′ and 5′-GTGGTGGTGGTGGTGCTCGATTTGGTTGCAATAAACAGACC-CCAACGC-3′; for *OsPEAMT2*, 5′-TAAGAAGGAGATATACATAATGGACGCCGTTGCAGCGAAC-3′ and 5′-GTGGTGGTGGTGGTGCTCGATTTGGTTGCGATAAACAGC-CCCCAAC-3′. The PCR products of *OsPEAMT1* and *OsPEAMT2* were cloned into the expression vector pET30a, respectively, and the expression plasmids of *OsPEAMT1* and *OsPEAMT2* were transformed into *E. coli* BL21 (DE3), respectively. The pET30a-*OsPEAMT1*-BL21 and pET30a-*OsPEAMT2*-BL21 were transferred to LB liquid medium containing kalamycin (40 μg/mL), respectively, and cultured at 37 °C until the OD reached 0.4–0.6. The recombinant OsPEAMT1 and OsPEAMT2 proteins were induced (10 h at 28 °C) by 0.2 mM Isopropyl β-D-1-thiogalactopyranoside (IPTG) and purified by Ni-NTA (His-tag) affinity chromatography [48]. Protein purity was estimated by SDS-PAGE and stained using Coomassie Brilliant Blue, and protein concentration was determined with Bradford reagent (Bio-Rad Laboratories, Inc., Hercules, CA, USA). The purified recombinant OsPEAMT1 and OsPEAMT2 proteins were stored at −80 °C.

### 4.5. Enzyme Assays of OsPEAMT1 and OsPEAMT2

Recombinant OsPEAMT1 and OsPEAMT2 protein activities were measured according to the published method [49] with some modifications; namely, preliminary experiments were conducted to screen the optimal temperature and pH for the enzymatic reactions of these two recombinant proteins. Then, a 1 mL assay mixture contained HEPES-KOH (pH 9.9), 2 mM EDTA, 200 μM P-EA (Sigma, St Louis, MO, USA), 200 μM S-adenosyl-L-methionine (Sigma), and 200 μL of OsPEAMT1 and OsPEAMT2 enzyme solution, respectively. The mixture was incubated for 30 min at 30 °C, and the reactions were stopped by shock-freezing in liquid N_2_. P-Cho, the enzymatic reaction product (P-Cho) of OsPEAMT1 and OsPEAMT2, was quantified using liquid chromatography tandem mass spectrometry (LC-MS/MS) as described below. For further determination of the activity of OsPEAMT1 and OsPEAMT2 in vivo, exogenous P-EA (0, 10, 20, and 40 mM) was added to 3-week-old rice seedlings. After two days of P-EA treatment, the P-Cho content of rice seedlings (three biological replicates per treatment) was quantified using LC-MS/MS, as described below.

### 4.6. Expression Analysis of OsPEAMT1 and OsPEAMT2

Total RNA was extracted from the rice leaves, roots, stems, flowers, and seeds at various developmental stages, and treated seedlings (salt or low-temperature treatment) were extracted using a modified CTAB (Cetyltrimethylammonium bromide) method [50]. The first strand of the reverse-transcribed cDNA was synthesized using the specifications of the Monad first-strand cDNA Synthesis Kit (Vazyme, Nanjing, China). For real-time fluorescence quantitative PCR (RT-qPCR), the rice actin gene (X16280, LOC112876597) was selected as a reference gene in this study because it is a conserved housekeeping gene and demonstrates relatively stable expression across different tissues and under various stress treatments in rice [29,51]. Forward and reverse primers for RT-qPCR analysis were designed for *OsACTIN* (forward, 5′-AGCTATCGTCCACAGGAA-3′; reverse, 5′-ACCGGAGCTAATCA GAGT-3′), *OsPEAMT1* (forward, 5′-GAGCACTGGTGGAATTGAAACT-3′; reverse, 5′-AGCACGCTCAAGTGCAAAAG-3′), and *OsPEAMT2* (forward, 5′-TGAAACCTGGGGGCAAAGTC-3′; reverse, 5′-TGAAGAACACGCAGGAACTGG-3′). The Tb Green^®^ Premix Ex Taq™ II (Takara, Beijing, China) was used for RT-qPCR, all data were obtained from three biological repetitions, and experiments were repeated three times. The RT-qPCR conditions were as follows: 95 °C for 30 s, followed by 40 cycles of 95 °C for 5 s, and 60 °C for 20 s. The relative expression level was determined by the 2^−ΔΔCt^ method [52].

### 4.7. Quantification of P-Cho

The quantification of P-Cho was determined by LC-MS/MS. The specific operations were as follows: for P-Cho extraction of samples, 50 mg of each individual sample was precisely weighed and transferred to a 2 mL Eppendorf tube. After the addition of 1000 μL of extract solvent (precooled at −20 °C, acetonitrile–methanol–water, 2:2:1), the samples were vortexed for 30 s, homogenized at 38 Hz for 4 min, and sonicated for 5 min in an ice-water bath, followed by incubation at −20 °C for one hour and centrifugation at 12,000 rpm and 4 °C for 15 min. A 50 μL aliquot of the clear supernatant was transferred to an auto-sampler vial for LC-MS/MS analysis. Another 10 μL of supernatant was diluted 10 times, 100 times, 1000 times for LC-MS/MS analysis. For the extraction of P-Cho from reaction solutions of OsPEAMT1 and OsPEAMT2 enzymes (see above), a 500 μL aliquot of each reaction solution was transferred to a 1.5 mL Eppendorf tube. After the addition of acetonitrile, the samples were vortexed for 30 s, followed by incubation at −20 °C overnight and centrifugation at 12,000 rpm and 4 °C for 15 min. A 600 μL aliquot of the clear supernatant was transferred to a new Eppendorf tube and dried under a gentle nitrogen flow. The residual was reconstructed with 180 μL of extraction solution (acetonitrile: methanol: water = 2:2:1, V: V: V), centrifuged at 12,000 rpm and 4 °C for 15 min. An 80 μL aliquot of the clear supernatant was transferred to an auto-sampler vial for LC-MS/MS analysis.

The UHPLC separation was carried out using a Waters ACQUITY H-class plus UPLC System, equipped with Agilent ZORBAX Eclipse Plus C18(Agilent Technologies, Shanghai, China) (2.1 mm × 150 mm, 1.8 μm). Mobile phase A was 0.1% formic acid and ammonium formate in water, and mobile phase B was methanol. The elution gradient is shown in Appendix A. The flow rate was 300 μL/min. The column temperature was set at 40 °C. The auto-sampler temperature was set at 10 °C, and the injection volume was 1 μL. A Waters Xevo TQ-XS triple quadrupole mass spectrometer, equipped with an electrospray ionization (ESI) interface, was applied for assay development. Typical ion source parameters were capillary voltage = −2500 V, cone voltage = 30 V, desolvation temperature = 550 °C, desolvation gas flow = 1000 L/Hr, collision gas flow = 0.15 mL/min, and nebulizer gas flow = 7 Bar. The MRM parameters for each of the targeted analytes were optimized by injecting the standard solutions of the individual analytes directly into the API source of the mass spectrometer. At least two MRM transitions (i.e., the Q1/Q3 pairs) per analyte were obtained, and the two most sensitive transitions were used in the MRM scan mode to optimize the collision energy for each Q1/Q3 pair. Among the two MRM transitions per analyte, the Q1/Q3 pairs that showed the highest sensitivity and selectivity were used as the MRM transitions for quantitative monitoring. The additional transitions acted as a qualifier for the purpose of verifying the identity of the target analytes. Waters MassLynx V4.2 Work Station Software was employed for MRM data acquisition and processing. The MRM parameters of the target compound are shown in Appendix A.

P-Cho standard was purchased from Sigma-Aldrich (Shanghai yuanye Bio-Technology Co., Ltd., Shanghai, China), and the stock solutions of P-Cho standard were prepared as a final concentration of 1 mmol/L. A 100 μL aliquot of each of the stock solutions was transferred to a 10 mL flask to form a mixed working standard solution. A series of calibration standard solutions were then prepared by stepwise dilution of this mixed standard solution. For calibration curves, the calibration solutions were subjected to UPLC-MRM-MS/MS analysis using the methods described above. The linear y = ax + b fittings of the calibration curves, where y is the ratio of peak areas for analyte, and x is the concentration (nmol/L) of the analyte, were used. The least squares method was used for the fittings. Next, 1/x weighting was applied in the curve fitting since it provided the highest accuracy and correlation coefficient (R^2^). The calibration curve is shown in Appendix A. The level was excluded from the calibration if S/N was close to or below 10, or the accuracy of calibration was not within 80–120%. The calibration standard solution was diluted stepwise, with a dilution factor of 2. These standard solutions were subjected to UHPLC-MRM-MS analysis. The signal-to-noise ratios were used to determine the limits of detection (LODs) and limits of quantitation (LOQs). The LODs and LOQs were defined as the analyte concentrations that led to peaks with signal-to-noise ratios (S/N) of 3 and 10, respectively, according to the US FDA guideline for bioanalytical method validation. The LODs were less than 19.53 nmol/L, and the LOQs were 19.53 nmol/L.

### 4.8. Analysis of Response of Exogenous P-Cho to Salt and Low-Temperature Stress in Rice

Exogenous P-Cho (0, 50, 100, and 150 mM) was applied to 3-week-old rice seedlings with similar growth. For low-temperature stress, these seedlings were placed in a growth chamber set at 10 °C for 16 h of light/8 h of dark. For salt stress, 150 mM NaCl was added to the seedlings that had been treated with different concentrations of exogenous P-Cho, and these seedlings were placed in a growth chamber set at 25 °C for 16 h of light/8 h of dark. After two days of low-temperature or salt stress, the phenotype of rice seedlings was observed and photographed. There were three biological replicates for each of the above treatments, and ten rice seedlings per biological replicate.

### 4.9. Statistical Analysis

The data are presented as mean ± standard error (SE). Mean comparison was performed using one-way analysis of variance (ANOVA), Duncan’s test, or Student’s *t* test in SPSS Statistics 27.0 (SPSS Inc., Chicago, IL, USA), and statistical significance was considered at *p* < 0.05. All figures were created using OriginPro 2023.

## 5. Conclusions

In this study, we conducted the first comprehensive functional dissection of the *PEAMT* gene family (OsPEAMT1 and OsPEAMT2) in rice. In vitro and in vivo experiments demonstrated that OsPEAMT1 and OsPEAMT2 can catalyze the conversion of P-EA to P-Cho, but the activity of OsPEAMT1 was significantly higher than that of OsPEAMT2 (*p* < 0.05). The two genes exhibited distinct expression patterns across various tissues and demonstrated divergent responses to salt and low-temperature stress. Additionally, we found that the exogenous P-Cho can increase the resistance of rice to low-temperature and salt stress. Our results validated the functional differentiation of *OsPEAMT1* and *OsPEAMT2*, elucidated their critical roles in rice stress resistance, and provided genetic resources and practical metabolites for crop stress resistance.

## Figures and Tables

**Figure 1 plants-14-02935-f001:**
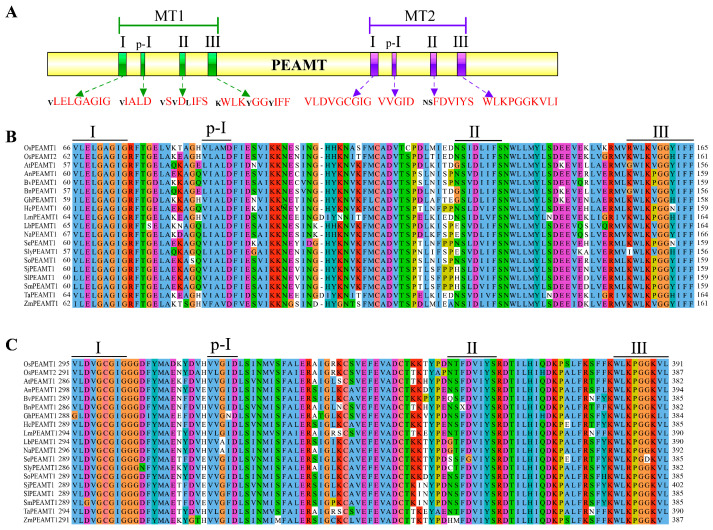
Multiple amino acid sequence alignment of OsPEAMT1 and OsPEAMT2. (**A**) The schematic diagram of plant PEAMTs. Methyltransferase domains MT1 and MT2 are shown in green and purple, respectively. The height of the letters is proportional to the occurrence frequency of the corresponding amino acids at the position. Black indicates that the sequence is generally consistent, and red indicates that the sequence is extremely consistent. (**B**,**C**) Multiple alignments of MT1 (**B**) and MT2 (**C**) domains. Dashes indicate the gaps introduced to improve the alignment. The same and similar amino acids are shown in a column in the same color. The I, p-I, II, and III of S-adenosyl-L-methionine-binding motifs are indicated. The protein accession numbers and their corresponding species are provided in the Appendix A.

**Figure 2 plants-14-02935-f002:**
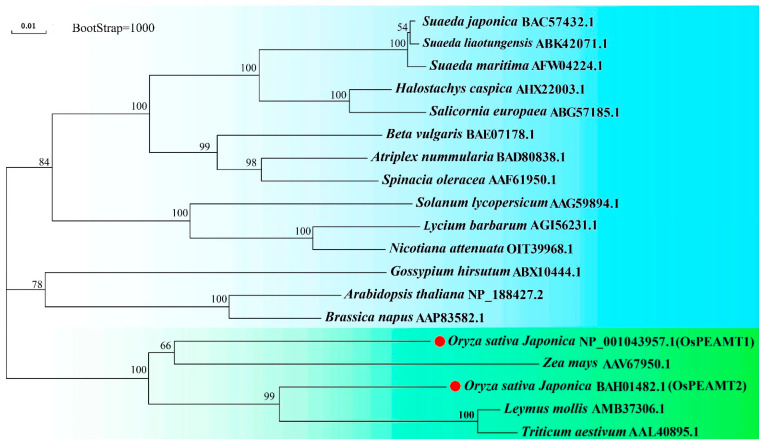
Phylogenetic analyses of plant PEAMTs. Phylogenetic analyses of plant PEAMT homologues using neighbor-joining method. The evolutionary distances were calculated using Poisson correction method. In total, 19 amino acid sequences of different plant PEAMTs are involved in this analysis. OsPEAMT1 and OsPEAMT2 are highlighted by red circles.

**Figure 3 plants-14-02935-f003:**
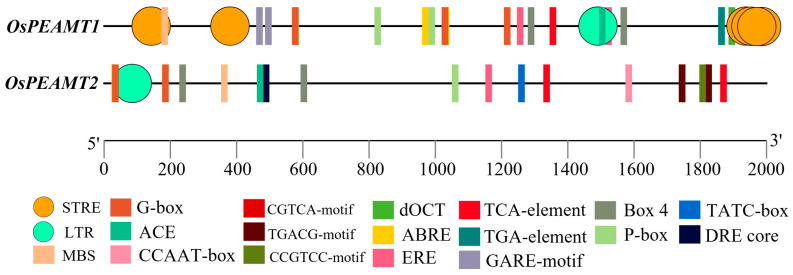
Distribution of *cis*-acting elements of *OsPEAMT1* and *OsPEAMT2* promoters. Different colors and shapes represent different *cis*-acting elements of *OsPEAMT1* and *OsPEAMT2* promoters. The definitions and implications of cis-acting element nomenclature are provided in the Appendix A.

**Figure 4 plants-14-02935-f004:**
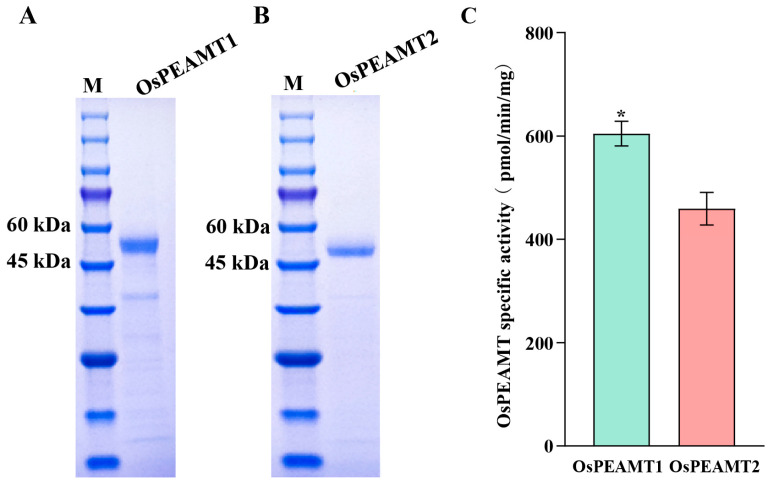
Functional analysis of the recombinant OsPEAMT1 (54.89 kD) and OsPEAMT2 (54.45 kD). (**A**,**B**), SDS-PAGE of purified recombinant OsPEAMT1 (**A**) and OsPEAMT2 proteins (**B**). Aliquots of purified protein (1 μg per lane) were electrophoresed. M, protein marker (Sigma-Aldrich, St Louis, MO, USA). (**C**) The specific activity of OsPEAMT1 and OsPEAMT2 (P-EA as the substrate). The enzyme activity determination is described in Materials and Methods. Values are means ± SE (n = 3 biological replicates). * *p* < 0.05 compared with each other, using Student’s *t* test.

**Figure 5 plants-14-02935-f005:**
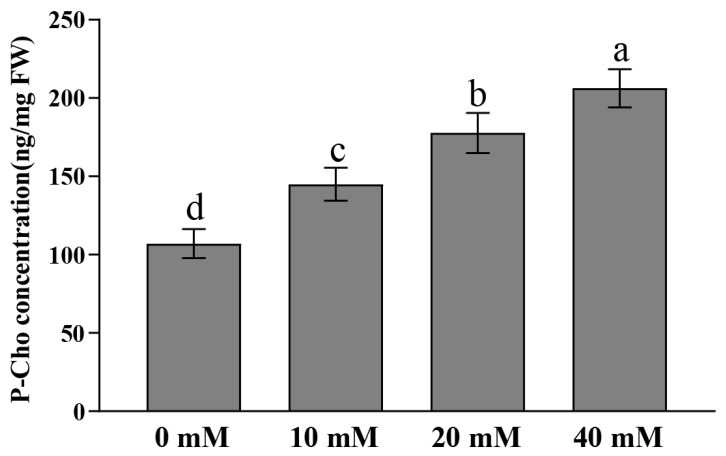
The determination of P-Cho after two days of exogenous substrate P-EA treatment in rice. The exogenous substrate P-EA (0, 10, 20, and 40 mM) was supplied to 3-week-old rice seedlings, respectively. P-Cho was determined by LC-MS/MS as described in Materials and Methods. Values are means ± SE of three biological replicates, and each replicate corresponds to the ten individual rice seedlings. One-way analysis of variance (ANOVA) followed by Duncan’s multiple comparison test were performed; different letters above the bars denote statistically significant differences (*p* < 0.05). FW: Fresh weight.

**Figure 6 plants-14-02935-f006:**
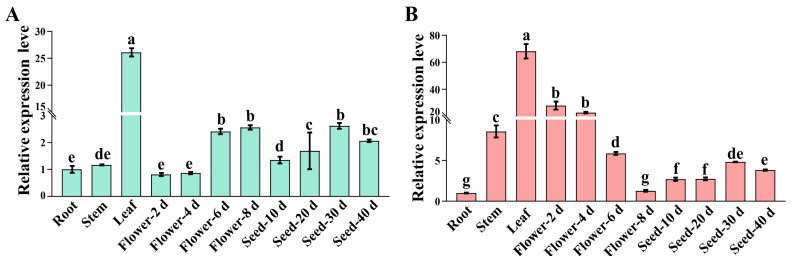
Transcript abundance of *OsPEAMT1* (**A**) and *OsPEAMT2* (**B**) in roots, stems, leaves, flowers, and seeds at various developmental stages. Data shown are means ± SE of three biological replicates, with each replicate corresponding to three individual plants. The roots, stems, and leaves were sampled from soil-grown plants (planted for 45 days) in the field. The flowers were sampled (n = 30) 2, 4, 6, and 8 days after rice flowering in the field. The seeds were sampled (n = 30) 10, 20, 30, and 40 days after rice grain filling in the field. One-way analysis of variance (ANOVA) followed by Duncan’s multiple comparison test were performed; different letters above the bars denote statistically significant differences (*p* < 0.05).

**Figure 7 plants-14-02935-f007:**
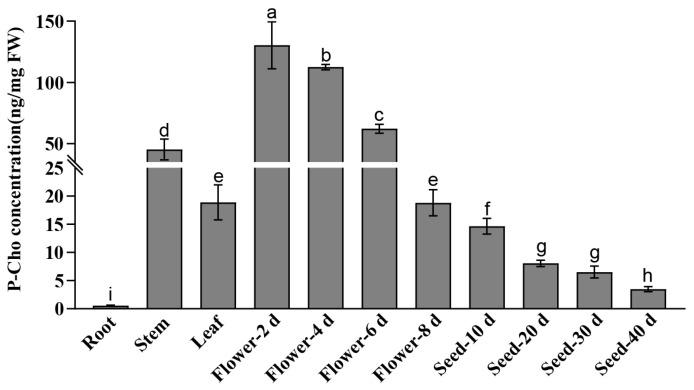
The determination of P-Cho in roots, stems, leaves, flowers, and seeds at various developmental stages. Data shown are means ± SE of three biological replicates, with each replicate corresponding to three individual plants. One-way analysis of variance (ANOVA) followed by Duncan’s multiple comparison test were performed; different letters above the bars denote statistically significant differences (*p* < 0.05). The sampling time and the number of samples were the same for the identification of *OsPEAMT1* and *OsPEAMT2* expression patterns. FW: Fresh weight.

**Figure 8 plants-14-02935-f008:**
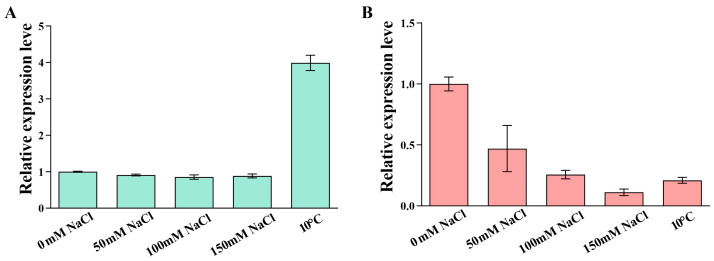
The expression pattern analysis of *OsPEAMT1* (**A**) and *OsPEAMT2* (**B**) in leaves in response to different concentrations of salt (0, 50, 100, and 150 mM) or low-temperature (10 °C) stress. Data shown are means ± SE of three biological replicates, with each replicate corresponding to ten individual plants. The leaves sampled from 3-week-old rice seedlings were treated with different salt concentrations or low temperatures for one day.

**Figure 9 plants-14-02935-f009:**
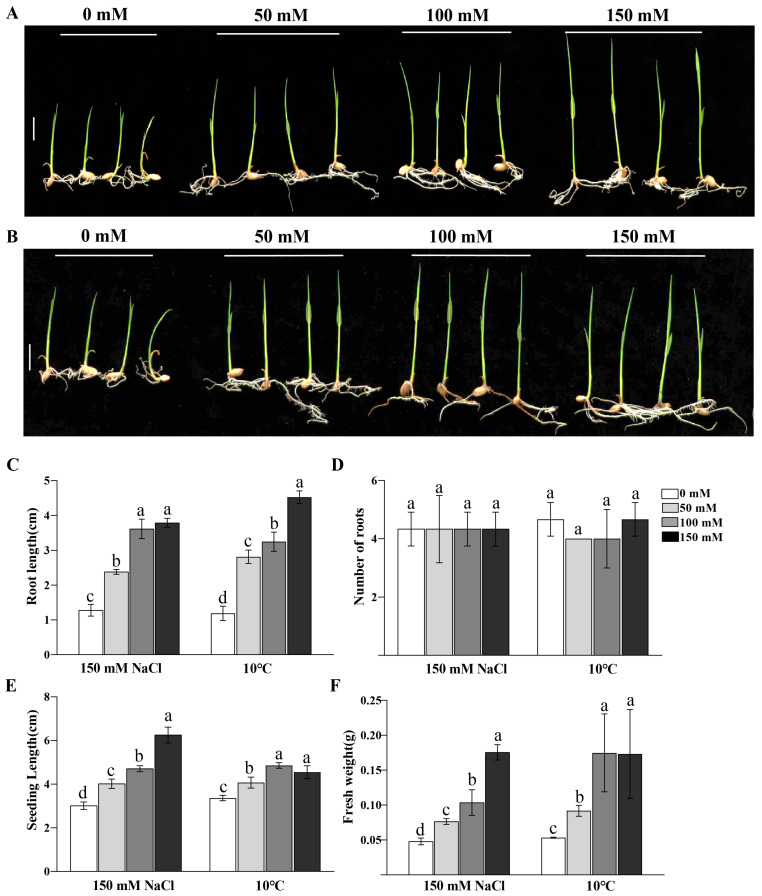
Exogenous P-Cho can enhance rice resistance to salt and low-temperature stress. (**A**,**B**) Representative photographs of rice seedlings under 150 mM NaCl (**A**) and 10 °C (**B**) stress after different concentrations of exogenous P-Cho were supplied. From left to right: 0, 50, 100, and 150 mM of exogenous P-Cho. Two-week-old rice seedlings with relatively uniform growth were transferred to bottles containing different concentrations of P-Cho, and some of these bottles were placed in a growth chamber set at 10 °C for 16 h of light/8 h of dark. In some of these bottles, 150 mM NaCl was added, and they were placed in a growth chamber set at 25 °C for 16 h of light/8 h of dark (as described in Materials and Methods). After 150 mM NaCl or 10 °C treatment for 3 days, the rice seedlings were taken out and photographed, and the root length (**C**), number of roots (**D**), seedling length (**E**), and fresh weight (**F**) were measured. Values are means ± SE of three biological replicates, with each replicate corresponding to the ten individual rice seedlings. One-way analysis of variance (ANOVA) followed by Duncan’s multiple comparison test were performed; different letters above the bars denote statistically significant differences (*p* < 0.05). Scale bar = 1 cm (**A**,**B**).

## Data Availability

The datasets used and/or analyzed in the current study are available from the corresponding author on reasonable request.

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
