# Peer review of "A Comprehensive Functional Analysis of OsPEAMT1 and OsPEAMT2 Genes in Rice (Oryza sativa L. ssp. japonica)"

_plants, 2025, doi:10.3390/plants14182935_

Round 1

Reviewer 1 Report

Comments and Suggestions for Authors

The manuscript provides a comprehensive functional analysis of OsPEAMT1 and OsPEAMT2 genes in rice, combining in silico, gene expression, enzymatic activity assays, and stress-response experiments. The work is relevant, the data are robust, and it has the potential to significantly contribute to the understanding of phosphatidylcholine metabolism and abiotic stress tolerance in cereals.

However, there are important points to improve.

L20–24: Briefly indicate how the analyses were performed (in silico, in vivo, and in vitro) so that the reader has an overview in the abstract.

L26–30: Avoid repeating "OsPEAMT1 and OsPEAMT2" multiple times in succession; they can be grouped as “both genes.”

L31–33: End with an impactful sentence linking the study to applications in molecular breeding.

L35–49: The introduction is detailed but contains redundancies. Condense the description of PC and GB functions, focusing on PEAMT’s role.

L50–75: Update references to include more recent works (2020–2025) on PEAMT in rice and other cereals.

L76–91: Improve the transition between the gene’s importance and the study’s rationale; currently, the justification appears fragmented.

L94–103: When describing protein size and molecular mass, also indicate the predicted isoelectric point (pI), as it is relevant for protein characterization.

L115–135 (Figure 1): The figure legend is excessively long and descriptive. Simplify and move details to the Supplementary Material.

L155–164: The cis-element results are presented visually, but the textual interpretation is superficial. Discuss possible biological implications.

L169–182 (Figure 3): The figure legend is excessively long and descriptive. Simplify and move details to the Supplementary Material.

L211–216: Better explain variation among concentrations and whether P-EA response reached saturation.

L245-257 (Figure 6): Add more visible error bars in figures and clearly indicate the statistical test used in the text.

L259–270: The discrepancy between P-Cho content and gene expression could be better discussed here, including plausible hypotheses.

L286–295: Include numerical or fold-change values in the text, not just qualitative interpretation.

L316: Correct “Figgure”.

L309–320: Standardize nomenclature (“low temperature” vs. “low-temperature”).

L385–401: Avoid unnecessary repetition of results already described; focus on biological interpretation and comparison with previous studies.

L416–425: Discuss in more depth possible mechanisms for P-Cho conversion into other metabolites, citing known metabolic pathways.

L430–442: Propose hypotheses for OsPEAMT2 downregulation under salinity—e.g., transcriptional repression by stress-related factors.

L443–452: Expand discussion on the potential use of exogenous P-Cho in agronomic management and rice breeding for stress tolerance.

L454–462: Specify planting density and environmental conditions (light, relative humidity) for field cultivation.

L479–502: Include antibiotic concentrations and induction time for heterologous expression.

L503–515: Indicate whether pH and temperature were optimized for each enzyme.

L516–531: Detail criteria for selecting reference genes in RT-qPCR.

L532–576: State detection (LOD) and quantification (LOQ) limits obtained experimentally.

L600–612: In concluson, rewrite more concisely, highlighting originality, main findings, and practical implications.

Some figures (especially Fig. 2, 3, and 8) have small fonts and poorly contrasting colors. Increase resolution and contrast.

This manuscript addresses an important topic and presents relevant findings; however, substantial revisions are necessary to improve clarity, methodological detail, and figure presentation.

Comments on the Quality of English Language

The English is functional, but the manuscript would benefit from a professional scientific language edit to ensure clarity, precision, and consistency suitable for publication.

Reviewer 2 Report

Comments and Suggestions for Authors

In this work, the authors cloned, expressed and biochemically characterized recombinant OsPEAMT1 and OsPEAMT2. In addition, they mapped tissue-specific and stress-responsive expression of both genes and quantified endogenous P-Cho. Moreover, they tested whether exogenous P-Cho enhances rice tolerance to salt and low-temperature stress. This work reports good results and manuscript is prepared well. I have the following comments and suggestions.

Section 2.7 and Fig. 9 mention 150 mM NaCl, whereas the Methods (L588-590) state 100 mM; reconcile or state the reason for the shift.

The assumption that externally supplied P-Cho enters rice tissues is not validated (no ¹³C-labelling, no tissue P-Cho quantification after feeding). Provide uptake data or qualify the claim.

Retention time, internal standard, and two MRM transitions (Q1/Q3 pairs) for P-Cho are missing

L596-598: “one-way ANOVA or Student’s t tests” is vague. State which test was used for each figure and report post-hoc tests (e.g., Tukey) when multiple comparisons are made

Can authors explain it. I might be wrong. The text claims “two methyltransferase domains” per protein, but the alignment (Fig. 1) shows one N-terminal MT domain and one C-terminal MT domain; clarify whether each domain is catalytically active for all three methylations or only partial steps.

I suggest authors add bootstrap values on Fig. 2 branches

The assertion that OsPEAMT2 is “negatively regulated” by salt/cold is based solely on transcript abundance. Provide evidence (e.g., promoter-reporter assays, ChIP-qPCR) that the observed decrease is transcriptional rather than post-transcriptional degradation; otherwise rephrase as “transcript levels decreased”.

“Figgure 9C” → “Figure 9C”.

Round 2

Reviewer 1 Report

Comments and Suggestions for Authors

The authors have addressed all the comments and implemented the requested corrections, which have improved the overall quality, clarity, and consistency of the work.